# Vitamin D Deficiency and Risk of Gestational Diabetes Mellitus in Western Countries: A Scoping Review

**DOI:** 10.3390/nu17152429

**Published:** 2025-07-25

**Authors:** Paola Correa, Hirukshi Bennett, Nancy Jemutai, Fahad Hanna

**Affiliations:** Program of Public Health, Department of Health and Education, Torrens University Australia, Melbourne VIC 3000, Australia; paola.correa@health.torrens.edu.au (P.C.); hirukshi.bennettrathnayake@torrens.edu.au (H.B.); nancy.jemutai@student.torrens.edu.au (N.J.)

**Keywords:** vitamin D, 25-hydroxyvitamin D, gestational diabetes mellitus, pregnancy, body mass index, maternal age, ethnicity, Western countries

## Abstract

**Background:** Gestational diabetes mellitus (GDM) is a prevalent pregnancy complication globally. Maternal vitamin D deficiency has been linked to the risk of GDM. The aim of this study was to explore and synthesise current evidence on the association between vitamin D deficiency and the development of gestational diabetes in Western countries. **Methods:** A scoping review was conducted in accordance with the Joanna Briggs Institute (JBI) methodological framework. Relevant studies were identified through a comprehensive search across seven databases: ProQuest Public Health, Google Scholar, PubMed, ScienceDirect, *The Lancet*, BMC Public Health, the *International Journal of Women’s Health*, and Scopus. Studies were included based on predefined inclusion and exclusion criteria relevant to the research question. The review followed the JBI protocol, and the PRISMA flowchart was used to guide and visualise the study selection process. **Results:** Nineteen studies were included in the final analysis, comprising research predominantly from Australia (5), the United States (5), and Canada (4). The findings indicate a notable association between vitamin D deficiency and GDM risk, moderated by factors such as maternal age, ethnicity, seasonal variation, and body mass index (BMI). Older maternal age and higher BMI were linked with lower vitamin D levels and a higher incidence of GDM. Ethnic groups with darker skin tones showed higher rates of vitamin D deficiency, increasing vulnerability to GDM. Seasonal patterns revealed lower vitamin D levels during winter months, correlating with greater GDM risk. These patterns underscore the need for targeted preventive strategies, including the potential role of vitamin D supplementation. **Conclusions:** This review supports an observed association between maternal vitamin D deficiency and increased GDM risk, influenced by demographic and environmental factors. While the evidence points to a potential preventative role for vitamin D, further high-quality research, including systematic reviews and meta-analyses, is essential to establish causality and inform clinical guidelines. The review identifies knowledge gaps and suggests directions for future research and clinical practice.

## 1. Introduction

Gestational diabetes mellitus (GDM) has become a highly prevalent disease, with 15–25% of pregnancies affected worldwide [1,2,3,4,5,6,7]. According to Rodriguez and Mahdy [8], GDM is defined as glucose intolerance at any level identified first during pregnancy, and GDM can be classified as A1GDM and A2GDM. A1GDM is gestational diabetes that is responsive to a controlled diet and managed without medication, whereas A2GDM is gestational diabetes managed with medication for optimal glycaemic control. Zhang et al. [9] state that GDM is considered a public health concern due to its adverse outcomes for both the mother and the offspring. GDM predisposes the mother and the offspring towards an increased risk of obesity and glucose intolerance in the future [10]. Consequently, it plays a key role in the public health issues of diabetes and obesity.

Furthermore, as Zhu and Zhang [11] point out, the prevalence of GDM varies significantly across different regions worldwide. Their research highlights higher rates of GDM in the Middle East, North Africa, Southeast Asia, and the Western Pacific, as well as lower rates in Europe. However, the authors also stress the complexity of directly comparing these rates due to the diverse population characteristics, screening methods, and diagnostic criteria used in different countries. This complexity underscores the need for a nuanced, global approach to understanding and addressing GDM.

Vitamin D plays a crucial role in maintaining the well-being of both mothers and foetuses during pregnancy [12,13]. The foetus transfers metabolite 25(OH)D from the bloodstream via the placenta [14]. Therefore, maintaining adequate levels of vitamin D in the mother is vital for the foetus’s health. However, vitamin D deficiency is a common issue among pregnant women, globally influenced by factors such as latitude, ethnicity, sunlight exposure, dietary intake, and skin pigmentation [15,16].

An in-depth analysis by Palacios and Gonzalez [17] has revealed a concerning trend of low serum vitamin D levels among pregnant and lactating women worldwide. The study found high rates of vitamin D deficiency in Asia, with rates as high as 38–41% in Kuwait, and in some high-income European countries, including Spain (20%), Canada (24%), America (33%), UK (35%), Netherlands (44%), Belgium (45%), and Germany (77%). These findings suggest a potential global inadequacy of vitamin D consumption, particularly during pregnancy, which is a serious public health concern.

Recently, interest has rapidly grown in understanding the association between vitamin D deficiency and the risk of gestational diabetes mellitus [18]. Some studies showed that Vitamin D deficiency causes increased insulin resistance in pregnant women [19,20,21]. Furthermore, Burris and Camargo Jr [22] reported that vitamin D intake by dietary supplementation could elevate levels of 25-hydroxyvitamin D, improving or preventing pregnancy complications caused by GDM. Moreover, a randomised placebo-controlled trial conducted by Shahgheibi et al. [23] also reported that taking Vitamin D supplements in the first and second trimesters of pregnancy contributed to controlled glucose levels and reduced the risk of GDM.

The aim of this study was to explore the impact of vitamin D deficiency in Western countries and how its deficiency can affect pregnant women developing gestational diabetes mellitus. Western countries were selected due to their shared health system structures, consistent diagnostic criteria for GDM, and high prevalence of vitamin D deficiency despite generally high-resource settings, allowing for more comparable data. A scoping review approach provides a methodology to identify evidence-based literature on a topic that is especially useful where issues require clarification before studies are conducted. Therefore, this study aims to analyse evidence-based research about the relationship between vitamin D and gestational diabetes in Western countries.

## 2. Research Design and Methodology

Reviewing existing literature is essential for enhancing health and clinical understanding. Two key approaches for analysing literature are systematic reviews and scoping reviews, each serving distinct purposes and following different methodologies. Systematic reviews focus on answering specific clinical questions and providing evidence to guide practice. In contrast, scoping reviews take a broader approach, identifying knowledge gaps and clarifying key concepts. The choice between these methods depends on the research objectives and the required depth of analysis [24,25]. A scoping review was chosen over a systematic review to broadly map the existing literature, identify knowledge gaps, and explore the range and nature of evidence available on the association between vitamin D deficiency and GDM, without restricting to narrowly defined outcomes or study types.

A comprehensive scoping literature review was conducted according to the “Joanna Briggs Institute (JBI) methodology for scoping review” [24,26]. The scoping review adopts an expanded research strategy to ensure transparency, reproducibility and reliability of existing knowledge in the field. This scoping review complied with the guidelines of Preferred Reporting Items for Systematic Reviews and Meta-Analyses-Extensions for Scoping Reviews (PRISMA-ScR) statement [25].

A structured search strategy was developed based on the JBI guidelines. The following keywords and Boolean operators were used across all databases:(“Vitamin D” OR “25-hydroxyvitamin D”) AND(“Gestational diabetes mellitus” OR “GDM”) AND(“Pregnan” OR “pregnancy”) AND *(“Western countr” OR “Australia” OR “Canada” OR “United States” OR “Europe”) *

Synonyms and Medical Subject Headings (MeSH) terms were adapted for each database. The search was conducted on 10 March 2025, across *ProQuest Public Health*, *Google Scholar*, *PubMed*, *Science Direct*, *The Lancet*, *BMC Public Health*, *International Journal of Women’s Health*, and *Scopus*.

Eligibility Criteria:

Inclusion criteria:Studies conducted in Western countries;Peer-reviewed articles examining the relationship between vitamin D deficiency and GDM;Observational or interventional designs;Published in English.

Exclusion criteria:Studies involving non-pregnant populations or non-Western settings;Reviews without original data;Studies lacking serum 25(OH)D measures or GDM outcomes;Conference abstracts, editorials, and grey literature.

### 2.1. Identification of Relevant Studies

The literature search was conducted from seven databases: ProQuest Public Health, Google Scholar, PubMed, Science Direct, *The Lancet*, BMC Public Health, *International Journal of Women’s Health*, and Scopus. The study utilised literature published in English and filtered it as peer-reviewed articles published between 2008 and 2024. The terms “Vitamin D” and “25-hydroxyvitamin” were selected because there are articles that opt to use the scientific form of vitamin D (25-hydroxyvitamin D) in their article title. Therefore, it would allow broader results when searching the literature. Similarly, using an asterisk on “pregnan” and “Western countr” permitted the database to find similar words for the search. For example, “pregnant/pregnancy” and “Western country/Western countries”. Utilising multiple databases and the keyword strategy allowed a more refined search related to the topic, which was helpful during the final appraisal [27].

### 2.2. Study Selection Criteria

Articles were eligible for inclusion in this scoping review after meeting the criteria of describing the relationship between vitamin D deficiency and gestational diabetes in Western countries. Moreover, only articles published between 2008 and 2024, articles written in English, and articles from any Western countries were eligible for inclusion in this study. The exclusion criteria were established for articles that did not describe the relationship between vitamin D and GDM and/or did not mention any Western country.

The titles and abstracts of peer-reviewed articles were screened first against the inclusion criteria mentioned above. Then, full texts were reviewed from the resulting articles for inclusion. Additional articles relevant to the topic were searched and included in the reference list.

### 2.3. Data Extraction

The study found 780 articles; however, 31 were excluded from the study due to not meeting the selection criteria. Of the remaining 749 articles, 590 were excluded after reading the title and abstract. In total, 159 articles were then assessed for eligibility, where 140 studies were excluded due to different populations, regions, and missing information. Ultimately, only 19 studies were selected for inclusion in this review. The PRISMA chart was followed to describe the search strategy (please refer to Figure 1). Data extraction was then performed based on standardised criteria for the scoping review. The included articles were distributed as follows: five from Australia, five from the United States, four from Canada, and one each from Norway, Italy, Spain, France, and the Netherlands. The methodology utilised in each article included nine cohort studies, four case–control studies, one double-blind randomised trial, two cross-sectional studies, three systematic reviews, and a meta-analysis (please refer to Table 1 for types of studies).

Braun and Clarke’s approach to thematic analysis was utilised to evaluate the data gathered from the data extraction [28]. The steps that comprised this approach were being familiar with the data, producing initial codes for the data, searching for potential themes, revising themes, defining and naming these, and reporting and analysing themes. PRISMA-ScR guidelines were used to report and analyse themes [25].

**Table 1 nutrients-17-02429-t001:** Characteristics of included studies.

Articles	Methodology	Title	Country	Adjustments	Summary of Findings	Study Subjects
Zhang et al. [29]	Cohort	Maternal Plasma 25-Hydroxyvitamin D Concentrations and the Risk for Gestational Diabetes Mellitus	United States	Maternal age, parity, ethnicity, season	Women classified as being deficient for vitamin D had a 3.7-fold increased subsequent risk of GDM, as compared with vitamin D-sufficient women.	953 women
Clifton-Bligh et al. [30]	Cohort	Maternal vitamin D deficiency, ethnicity, and gestational diabetes	Australia	Age, weight, height, ethnicity	The odds ratio of gestational diabetes in women with 25OHD < 50 nmol/L did not reach statistical significance.	307 women
Lau et al. [31]	Retrospective Cross-sectional Study	Serum 25-hydroxyvitamin D and glycated haemoglobin levels in women with gestational diabetes mellitus	Australia	Ethnicity, weather season, occupational status, age, BMI, fasting glucose level	Ethnicity was associated with 25(OH)D levels and HbA1c levels in pregnant women with GDM. Season was also associated with 25(OH)D levels and HbA1c levels in women with GDM where spring showed the highest association, summer showed the lowest and autumn and winter the average.	147 women
Parlea et al. [32]	Case–Control Study	Association between serum 25-hydroxyvitamin D in early pregnancy and risk of gestational diabetes mellitus	Canada	Age, gestational age, weight, race, season	There was a significant association between vitamin D deficiency and risk of gestational diabetes after adjusting for gestational age and maternal weight. The associations between vitamin D deficiency and GDM were also related to weather season where summer showed higher results compared with winter months.	116 women
Poel et al. [33]	Systematic Review and Meta-analysis	Vitamin D and gestational diabetes: A systematic review and meta-analysis	Netherlands	Age, BMI, ethnicity	Association appeared stronger among Caucasian compared with non-Caucasian women. Even when adjusted ethnicity as confounding factor, the association between vitamin D and GDM remained significant statistically. When adjusted the results for BMI and maternal age, the association between vitamin D and GDM remained significant.	7 studies
Burris et al. [22]	Cohort Study	Vitamin D Deficiency in Pregnancy and Gestational Diabetes	United States	Maternal age, season, gestational age, ethnicity, marital status, smoking, parity	Second trimester 25(OH)D levels were inversely associated with glucose levels after 1 h 50 g glucose challenge test and low 25(OH)D levels may be associated with increased risk of GDM	1314 women
Tomedi et al. [34]	Cohort	Early-pregnancy maternal vitamin D status and maternal hyperglycaemia	United States	Gestational age, ethnicity, age, parity, BMI	Each 23-nmol/L increase in serum 25-hydroxyvitamin D was associated with a reduction in the odds of maternal hyperglycaemia.	429 women
Wei et al. [35]	Systematic Review and Meta-analysis	Maternal vitamin D status and adverse pregnancy outcomes: a systematic review and meta-analysis	Canada	None	Women with circulating 25-hydroxyvitamin D [25(OH)D] level less than 50 nmol/L in pregnancy experienced an increased risk of GDM.	24 studies
Lacroix et al. [36]	Cohort	Lower vitamin D levels at first trimester are associated with higher risk of developing gestational diabetes mellitus	Italy	Age, ethnicity, BMI, season, vitamin d supplementation	Lower first trimester 25OHD levels were associated with an increased risk of developing GDM during pregnancy.	655 women
Yap et al. [37]	Double-blind Randomised Control Trial	Vitamin D Supplementation and the Effects on Glucose Metabolism During Pregnancy: A Randomized Controlled Trial	Australia	Age, BMI, ethnicity, vitamin d supplementation	Commencing high dose vitamin D supplementation at 14 weeks gestation did not improve maternal glucose levels in pregnancy however maternal baseline vitamin D levels < 32 ng/mL, 5000 IU per day was highly effective in preventing neonatal vitamin D deficiency.	179 Women
Nobles et al. [38]	Prospective Study	Early pregnancy vitamin D status and risk for adverse maternal and infant outcomes in a bi-ethnic cohort: the Behaviors Affecting Baby and You (B.A.B.Y.) Study	United States	Age, gestational age, BMI, ethnicity	After accounting for factors such as the month and gestational age at blood draw, gestational age at delivery, age, BMI, and Hispanic ethnicity, women with insufficient and deficient vitamin D levels had infants with birth weights 139.74 g (SE 69.16, *p* = 0.045) and 175.52 g (SE 89.45, *p* = 0.051) lower than those with sufficient vitamin D levels (≥30 ng/mL). Among Hispanic women, each 1 ng/mL increase in 25(OH)D was associated with a higher risk of gestational diabetes mellitus (relative risk 1.07; 95% CI 1.03, 1.11). No significant associations were found between maternal vitamin D levels and other pregnancy outcomes.	237 women
Arnold et al. [39]	Nested Case–Cohort Study	Early Pregnancy Maternal Vitamin D Concentrations and Risk of Gestational Diabetes Mellitus	United States	Maternal age, ethnicity, season, family history of diabetes, BMI	Women in the lowest quartile for 25[OH]D3 concentration had higher risk of GDM compared with women in the highest quartile.	135 women
Dodds et al. [40]	Nested Case–Control Study	Vitamin D Status and Gestational Diabetes: Effect of Smoking Status during Pregnancy	Canada	Maternal age, BMI, season, gestational week, smoking	Pregnant women who smoked and had 25(OH)D levels < 30 nmol/L showed an adjusted odds ratio (aOR) of 3.73 (95% CI 1.95, 7.14) compared to non-smokers with 25(OH)D ≥ 50 nmol/L. Additive interaction between smoking and low 25(OH)D levels was observed (RERI = 2.44, 95% CI 0.03, 4.85). The findings highlight an inverse relationship between vitamin D levels and gestational diabetes risk, particularly among smokers.	395 women
Wilson et al. [41]	Cohort Study	Vitamin D levels in an Australian and New Zealand cohort and the association with pregnancy outcome	Australia and New Zealand	Ethnicity, age, solar exposure, geographical locations, genetics, supplementation	A 53% decreased risk for gestational diabetes mellitus (GDM) was observed with high vitamin D status when compared to moderate-high.	2800 women
Eggemoen et al. [42]	Cohort	Vitamin D, Gestational Diabetes, and Measures of Glucose Metabolism in a Population-Based Multiethnic Cohort	Norway	Age, parity, ethnicity, season	Maternal age strongly associated with the vitamin status and pregnant women diagnosed with GDM. After adjusting for prepregnant BMI and weight gain during pregnancy, vitamin D deficiency and GDM were still significantly associated.	745 women
Griew et al. [43]	Cohort Study	Early pregnancy vitamin D deficiency and gestational diabetes: Exploring the link	Australia	Family history, ethnicity, body mass index, age	Maternal age showed one of the key predictors of developing GDM. Ethnicity has strong association between vitamin D and GDM.	785 women
Sadeghian et al. [44]	Systematic Review and Dose–Response Meta-analysis	Circulating vitamin D and the risk of gestational diabetes: a systematic review and dose–response meta-analysis	Canada	BMI, gestational age, ethnicity, smoking status	The analysis included nine cohort studies and six nested case–control studies, comprising 40,788 participants and 1848 gestational diabetes mellitus (GDM) cases. In a linear analysis, each 10 nmol/L increase in circulating 25(OH)D was linked to a 2% reduction in GDM risk (effect size (ES): 0.98; 95% CI: 0.98, 0.99; *I*^2^ = 85.0%, *p* < 0.001). Comparing the highest to the lowest 25(OH)D levels showed a 29% reduced GDM risk with moderate heterogeneity (*I*^2^ = 45.0%, *p* = 0.079). In conclusion, lower serum 25(OH)D levels were associated with an increased GDM risk.	15 studies (9 cohort and 6 nested case–control studies)
Salakos et al. [45]	A Nested Case–Control Study	Relationship between vitamin D status in the first trimester of pregnancy and gestational diabetes mellitus—A nested case–control study	France	Age, BMI before pregnancy, conception season, parity	The GDM risk was significantly greater for patients with 25OHD levels < 20 ng/mL. However, there seems to be no linear relationship between GDM and 25OHD levels in the first trimester of pregnancy since GDM risk does not continuously decrease as 25OHD concentrations increase.	250 women
Agüero-Domenech et al. [46]	Cross-sectional Study	Vitamin D Deficiency and Gestational Diabetes Mellitus in Relation to Body Mass Index	Spain	BMI, age, ethnicity	Vitamin D deficiency is associated with gestational diabetes, but is independent of BMI.	886 women

### 2.4. Data Analysis

In the data analysis, we utilised a scoping review method. Scoping reviews have become popular over the years due to synthesised research evidence [47]. The scoping review process identifies the research question, defines inclusion and exclusion criteria, uses evidence-based search, and selects articles that have already been published. In our study, we extracted the evidence, charted the evidence, and presented the results by reviewing all content analyses regarding the findings of the relationship between vitamin D deficiency and GDM in Western countries.

## 3. Results

The database search yielded a total of 780 full-text articles, as illustrated in Figure 1. After removing 31 duplicates, 749 articles were screened by title and abstract, resulting in 590 exclusions based on predefined eligibility criteria. A total of 159 full-text articles were then assessed, of which 138 were excluded due to population mismatch (*n* = 40), studies conducted outside Western countries (*n* = 51), or lack of critical information (*n* = 47). This process resulted in 19 studies being included in the final review.

### 3.1. Study Characteristics

The included studies varied in methodology and scope:**Study types:** cohort studies (*n* = 7), cross-sectional (*n* = 4), retrospective analyses (*n* = 3), randomised controlled trials (*n* = 2), and systematic reviews with meta-analyses (*n* = 3).**Countries represented:** Australia (*n* = 5), United States (*n* = 5), Canada (*n* = 4), and other Western countries.**Vitamin D exposure measures:** serum 25(OH)D concentration was the primary biomarker used across all studies.**Outcome definition:** GDM was defined according to local or WHO diagnostic criteria in all included studies.

Five main thematic categories were identified:Association between vitamin D deficiency and GDM;Maternal age as a modifying factor;Ethnic variation in vitamin D status and GDM;Seasonal variation;Body mass index (BMI) as a contributing factor.

See Table 1 for detailed study characteristics.

### 3.2. Study Themes

#### 3.2.1. Association Between Vitamin D Deficiency and GDM in Observational Studies

Of the 19 studies reviewed, 13 found a significant association between vitamin D deficiency and the development of GDM, as shown below:Wilson et al. [41] found that higher serum vitamin D levels in early pregnancy reduced GDM risk (OR = 0.47; 95% CI: 0.23–0.96).Sadeghian et al. [44] reported a 2% reduction in GDM risk per 10 nmol/L increase in 25(OH)D.Burris et al. [22] showed an inverse relationship between second-trimester 25(OH)D levels and glucose challenge test results.Several other studies—including Zhang et al. [29], Lacroix et al. [36], Arnold et al. [39], Tomedi et al. [34], Agüero-Domenech et al. [46], Dodds et al. [40], and Wei et al. [35]—consistently supported this association, even after adjusting for confounders such as BMI, parity, and ethnicity.

However, some studies presented mixed findings. Yap et al. [37] observed that while high-dose vitamin D supplementation did not improve maternal glucose levels, it significantly reduced neonatal vitamin D deficiency. Likewise, Clifton-Bligh et al. [30] found no statistically significant association between vitamin D deficiency and GDM (OR = 1.92; 95% CI: 0.89–4.17).

#### 3.2.2. Theme 2: Maternal Age

Three studies explored maternal age as a potential modifier of the vitamin D–GDM relationship. Griew et al. [43] found maternal age to be a strong predictor of GDM (OR = 1.05; 95% CI: 1.01–1.09; *p* = 0.012). Similarly, Eggemoen et al. [42] and Parlea et al. [32] reported that older maternal age correlated with lower 25(OH)D levels and increased GDM risk.

#### 3.2.3. Theme 3: Ethnicity

Five studies explored ethnicity as a confounding or modifying factor in the vitamin D–GDM relationship:Griew et al. [43] found a stronger association in Middle Eastern and Caucasian women.Lau et al. [31] and Nobles et al. [38] reported lower vitamin D levels and worse pregnancy outcomes among ethnic minority groups, particularly those with darker skin pigmentation.Parlea et al. [32] observed a stronger association in Caucasians compared to non-Caucasians.Poel et al. [33] reinforced ethnicity as a significant factor influencing vitamin D status and GDM risk.

#### 3.2.4. Theme 4: Seasonal Variation

Two studies highlighted the role of seasonality in vitamin D status. Parlea et al. [32] reported higher GDM risk in summer (aOR = 4.13; 95% CI: 1.44–11.80) compared to winter. Lau et al. [31] found variations in 25(OH)D and HbA1c levels across seasons, with spring showing the strongest association.

#### 3.2.5. Theme 5: Body Mass Index (BMI)

Two studies evaluated BMI as a contributing factor. Poel et al. [33] and Eggemoen et al. [42] both found that even after adjusting for BMI, the association between vitamin D deficiency and GDM remained statistically significant, suggesting that obesity may exacerbate this relationship.

## 4. Discussion

This scoping review highlights a potential association between vitamin D deficiency and increased risk of gestational diabetes mellitus (GDM) in Western countries, influenced by factors such as maternal age, ethnicity, seasonal variation, and body mass index (BMI). These variables appear to either exacerbate vitamin D deficiency or magnify its impact on glucose metabolism and insulin regulation during pregnancy.

Several mechanisms support the biological plausibility of this association. Vitamin D, through its interaction with pancreatic β-cell receptors, may influence insulin secretion and overall glucose homeostasis [48]. Additionally, vitamin D contributes to insulin sensitivity by activating insulin receptors and maintaining intracellular calcium levels essential for insulin-mediated pathways [49,50]. These mechanisms suggest a critical role for vitamin D in maintaining metabolic stability during pregnancy.

However, findings across studies remain mixed. For instance, Zhang et al. [29] reported a 3.7-fold increased risk of GDM among vitamin D-deficient women, whereas Clifton-Bligh et al. [30] found no statistically significant association. These discrepancies may stem from variations in study design, such as small sample sizes, differing gestational timing of serum 25(OH)D measurement, or vitamin D supplementation protocols. Additionally, heterogeneity in GDM diagnostic criteria and unmeasured confounders (e.g., dietary intake, sun exposure) may have influenced effect estimates. These inconsistencies highlight the need for more standardised methodologies and underscore the importance of caution in drawing causal inferences from observational studies.

In alignment with the current review, Mithal and Kalra [51] observed that vitamin D deficiency may contribute to adverse pregnancy outcomes, including GDM, although the evidence supporting vitamin D supplementation as a preventive strategy remains limited. Nonetheless, several high-quality studies consistently identified maternal age, BMI, ethnicity, and seasonality as important moderators of the vitamin D–GDM relationship [52,53,54].

Further evidence suggests that vitamin D, being a fat-soluble compound, may become sequestered in adipose tissue, reducing its bioavailability in women with obesity [55]. Similarly, darker skin pigmentation and limited sunlight exposure—common among certain ethnic groups or during winter months—negatively influence endogenous vitamin D synthesis, further complicating its role in GDM risk.

From a public health perspective, this review supports the rationale for targeted antenatal interventions. Based on the accumulated evidence and existing guidelines, some clinical bodies recommend vitamin D supplementation during pregnancy, particularly in winter or among high-risk groups [56]. Interventions such as dietary fortification, routine antenatal vitamin D screening, and tailored education programs may represent feasible, low-cost strategies to reduce GDM risk.

Importantly, the COVID-19 pandemic added a new dimension to the issue. Post-COVID-19 studies found that factors such as lockdowns and restricted outdoor mobility significantly reduced sun exposure, which may have further contributed to widespread vitamin D insufficiency in pregnant populations [57]. Sinaci et al. [58] suggest that vitamin D supplementation could not only mitigate GDM risk but also potentially reduce adverse COVID-19 outcomes in pregnancy, underscoring its broader relevance in maternal health.

Finally, the observed disparities by ethnicity and socioeconomic status indicate a need for equity-focused interventions. Health literacy, access to prenatal care, and culturally appropriate education on vitamin D intake and sun exposure should be integrated into antenatal strategies. This calls for a multisectoral approach, combining public health policy, primary care, and patient education to address modifiable risk factors and reduce maternal health inequities.

This review is limited by its geographic focus on Western countries, which excluded potentially relevant evidence from high-GDM-burden regions such as Asia and the Middle East. We acknowledge that patterns of vitamin D deficiency and GDM may differ significantly in non-Western settings due to environmental, cultural, and healthcare system factors. Comparative analyses across diverse regions would offer valuable insights for future global research.

There was notable heterogeneity among the included studies in terms of study design, population characteristics, diagnostic criteria for GDM, and methods of measuring vitamin D status. The predominance of observational studies further limits causal interpretation. Nevertheless, the findings highlight associations that warrant further exploration, particularly given the feasibility and low risk of vitamin D-based preventive strategies.

Finally, language restrictions and the exclusion of grey literature may have led to the omission of relevant studies, potentially limiting the comprehensiveness of the evidence base.

Given the consistent association observed between maternal vitamin D deficiency and increased GDM risk, routine antenatal screening for vitamin D levels should be considered, particularly for high-risk groups, such as women with higher BMI, older maternal age, darker skin pigmentation, and those pregnant during winter months. Public health campaigns should promote vitamin D-rich diets, safe sun exposure, and culturally appropriate physical activity.

Healthcare systems and policymakers should evaluate the integration of vitamin D screening and supplementation into existing prenatal care protocols, particularly in populations with known deficiency prevalence. These strategies are low-cost, feasible, and potentially scalable. In parallel, further research, particularly large, longitudinal randomised controlled trials, is needed to confirm causality and establish optimal supplementation thresholds and timing.

## 5. Conclusions

This scoping review supports a growing body of evidence linking maternal vitamin D deficiency with an elevated risk of gestational diabetes mellitus (GDM), influenced by maternal age, ethnicity, BMI, and seasonal variation. While causality cannot be definitively established, the consistency of associations across multiple studies supports proactive clinical action.

In the absence of clear causal evidence, integrating targeted vitamin D screening and supplementation into antenatal care—particularly for high-risk subgroups—represents a practical and evidence-informed approach to reducing GDM risk. These findings highlight the broader need to embed maternal nutrition and metabolic health into antenatal policy and practice, with an emphasis on prevention, early identification, and equitable care delivery.

## Figures and Tables

**Figure 1 nutrients-17-02429-f001:**
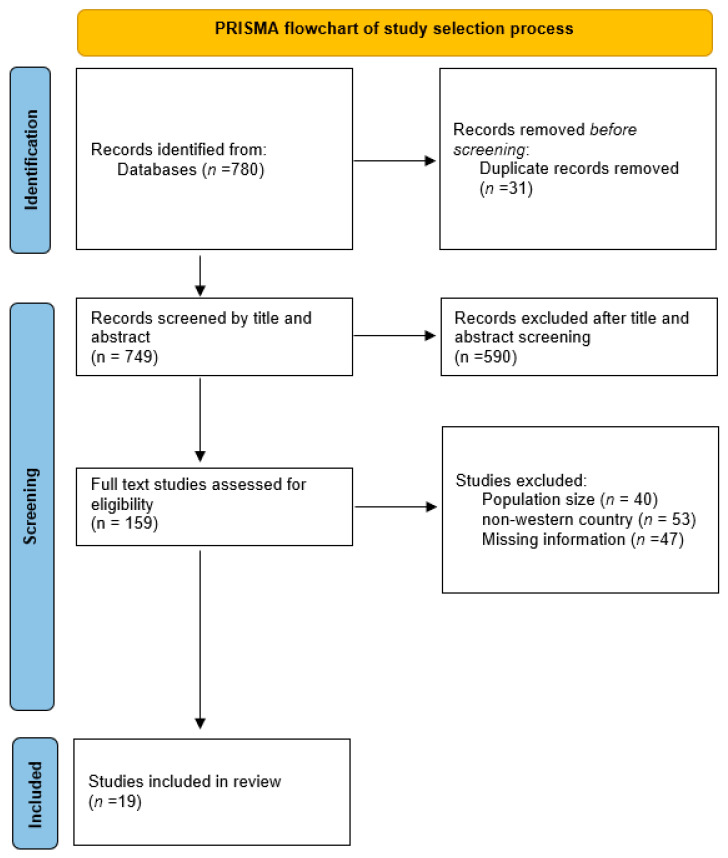
PRISMA flow diagram of included studies.

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
