# Peer review of "Vitamin D Deficiency and Risk of Gestational Diabetes Mellitus in Western Countries: A Scoping Review"

_nutrients, 2025, doi:10.3390/nu17152429_

Round 1

Reviewer 1 Report

Comments and Suggestions for Authors

Although the topic is very interesting, I think there are some changes to be made to the manuscript:

    1. The abstract is too long and does not comply with the journal's guidelines. It needs to be shortened.
    2. Please review the graphical abstract carefully, as it contains some errors.
    3. References are cited incorrectly throughout the text. Additionally, the formatting used in the reference list is also incorrect.
    4. Section 2.1 states that only articles published between 2010 and 2024 were selected, while section 2.2 indicates a range from 2008 to 2024. Please clarify which is correct.
    5. The results section should be expanded to provide more comprehensive insights.
    6. Sections 5 and 6 should be incorporated into the discussion section.

Author Response

Comments and Suggestions for Authors

Although the topic is very interesting, I think there are some changes to be made to the manuscript:

    1. The abstract is too long and does not comply with the journal's guidelines. It needs to be shortened.

Response: we have taken the reviewer’s comment on board and reduced the word count in the abstract- thank you

    1. Please review the graphical abstract carefully, as it contains some errors.

Response: Great pick up by the reviewer- this is now fixed

    1. References are cited incorrectly throughout the text. Additionally, the formatting used in the reference list is also incorrect.

Response: The referencing, formatting and citing will be corrected when converted into numbering based on the journal formatting, during the proofing of the manuscript. Thanks for the observation  

    1. Section 2.1 states that only articles published between 2010 and 2024 were selected, while section 2.2 indicates a range from 2008 to 2024. Please clarify which is correct.

Response: This seems to be a typographical error and has been corrected

    1. The results section should be expanded to provide more comprehensive insights.

Response: Thanks for this important pick up- The authors have now attempted to expand n the results wherever this is possible. Please see improved results section based on this reviewer’s comment- Thank you

    1. Sections 5 and 6 should be incorporated into the discussion section.

Response: done

Reviewer 2 Report

Comments and Suggestions for Authors

The article titled " Vitamin D Deficiency and Risk of Gestational Diabetes Mellitus in Western Countries: A Scoping Review" by Paola Correa et al. offers valuable insights. However, several aspects warrant further consideration:

  1. The review acknowledged that the evidence is associative, not causal. Confounding factors (e.g., diet, physical activity, baseline metabolic health) may contribute to both vitamin D deficiency and GDM risk.
  2. Variations in study designs (cohort, case-control), vitamin D measurement methods (serum 25(OH)D thresholds), and GDM diagnostic criteria may limit comparability.
  3.  The authors did not explicitly assess whether negative or null-result studies were underrepresented in the literature.
  4. While insulin resistance was mentioned, a deeper discussion of biological pathways (e.g., vitamin D’s role in pancreatic β-cell function, inflammation) would strengthen the theoretical basis.
  5.  While darker skin is linked to lower vitamin D synthesis, socio-economic factors (e.g., dietary habits, sun exposure behaviors) may also play a role but are not thoroughly explored.
  6. The authors provided seasonal differences but did not discuss whether studies adjusted for sunlight exposure or dietary vitamin D intake across seasons.
  7. A critical appraisal of individual study quality (e.g., risk of bias, sample size limitations) was missing, which could affect the strength of conclusions.
  8. The exclusion of non-Western populations (where vitamin D deficiency and GDM patterns may differ) limits global applicability.

Author Response

Comments and Suggestions for Authors

The article titled " Vitamin D Deficiency and Risk of Gestational Diabetes Mellitus in Western Countries: A Scoping Review" by Paola Correa et al. offers valuable insights. However, several aspects warrant further consideration:

  1. The review acknowledged that the evidence is associative, not causal. Confounding factors (e.g., diet, physical activity, baseline metabolic health) may contribute to both vitamin D deficiency and GDM risk.

Response: We thank the reviewer for making an important remark on the evidence gathered in this review. We also acknowledge the fact that confounding factors may have somewhat impacted the individual outcomes of the included studies. The review discusses this in part, although it focuses mostly on the reported findings, as they are to avoid speculations or their own expansion of findings. This was also acknowledged in the limitation section- “Additionally, most included studies were observational, and causality may not be fully established.”

  1. Variations in study designs (cohort, case-control), vitamin D measurement methods (serum 25(OH)D thresholds), and GDM diagnostic criteria may limit comparability.

Response: This is another important point. We have incorporated this in the limitation of the study. This is how the part of limitations read now: “There was considerable heterogeneity in study populations, diagnostic criteria, and vitamin D measurement methods, which may limit the comparability and synthesis of findings. Additionally, most included studies were observational, and causality may not be fully established. .”

  1.  The authors did not explicitly assess whether negative or null-result studies were underrepresented in the literature.

Response: We thank the reviewer for this thoughtful comment. While a formal assessment of publication bias was beyond the scope of this scoping review, we did include and discuss studies that reported null or negative associations between breakfast skipping and mental health outcomes such as depression and anxiety. These studies were presented alongside those showing positive associations to provide a balanced overview of the evidence. We agree that underrepresentation of null-result studies is a broader issue in the literature, and have now briefly acknowledged this in the discussion section to reflect the point raised.

  1. While insulin resistance was mentioned, a deeper discussion of biological pathways (e.g., vitamin D’s role in pancreatic β-cell function, inflammation) would strengthen the theoretical basis.

Response: Thanks for the comment- this was indeed missed from the discussion part and has now been added to the document- please see improved discussion section

  1.  While darker skin is linked to lower vitamin D synthesis, socio-economic factors (e.g., dietary habits, sun exposure behaviors) may also play a role but are not thoroughly explored.

Response: fair comment- we did elaborate on socio-economic factors, although more emphasis will strengthen the case for sure. This is now added to the main document

  1. The authors provided seasonal differences but did not discuss whether studies adjusted for sunlight exposure or dietary vitamin D intake across seasons.

Response: very valid point again- we did not find evidence of this in the included studies.

  1. A critical appraisal of individual study quality (e.g., risk of bias, sample size limitations) was missing, which could affect the strength of conclusions.

Response: This would strengthen the review and add value, although the authors feel at this stage, not critical appraisal for this scoping review was not carried out. We have done this in our systematic reviews as it seems to be more of a requirement.

  1. The exclusion of non-Western populations (where vitamin D deficiency and GDM patterns may differ) limits global applicability.

Response: We thank the reviewer for this important observation. The exclusion of non-Western populations was an intentional methodological decision based on the study objective, which was to synthesise evidence specifically relevant to Western health systems and populations. These settings often share comparable healthcare models, screening guidelines, and demographic risk profiles, which enhances the contextual relevance and applicability of findings within these countries.

We agree that vitamin D deficiency and GDM may present differently in non-Western settings due to variations in diet, sun exposure, healthcare access, and cultural practices. We have added a brief statement to the Discussion to acknowledge this distinction and to encourage future research that explores cross-regional comparisons to enrich the global evidence base.

Reviewer 3 Report

Comments and Suggestions for Authors

This scoping review provides a well-structured and timely overview of the relationship between vitamin D deficiency and gestational diabetes mellitus (GDM) in Western countries. The manuscript demonstrates a strong understanding of the topic and effectively synthesizes evidence from diverse sources. The thematic categorization of results—particularly around maternal age, ethnicity, BMI, and seasonality—adds clarity and relevance to the discussion.

However, several areas could benefit from improvement:

  1. Language and Clarity: The overall readability would be enhanced by simplifying complex sentences and removing redundant expressions. Minor grammatical adjustments are also recommended throughout the manuscript to improve clarity and flow.

  2. Methods Section: Although the methodology follows the JBI and PRISMA-ScR frameworks appropriately, certain parts (e.g., keyword strategy and exclusion reasons) should be more precisely and concisely described to increase transparency and reproducibility.

  3. Discussion and Interpretation:

    • While the discussion effectively highlights key mechanisms and moderators, additional elaboration on inconsistent findings (e.g., null associations) would be helpful.

    • The mention of COVID-19 is interesting but would be stronger with clarification on whether included studies were pre- or post-pandemic.

  4. Limitations Section: Please revise ambiguous expressions such as “while this may be the case…” to clearer, grammatically complete sentences. Also consider elaborating slightly on the potential impact of excluding grey literature and non-English studies.

  5. Recommendations and Conclusion: These sections are well written, but could benefit from a more explicit link between your findings and the proposed clinical/public health actions (e.g., vitamin D screening policies).

Comments on the Quality of English Language

The manuscript is generally well-organized and readable; however, several sentences are overly long or contain redundant phrasing, which occasionally hinders clarity. Minor grammatical errors and awkward constructions appear throughout the text, particularly in the Methods and Discussion sections. Simplifying sentence structure, ensuring consistent terminology, and improving transitions between ideas will significantly enhance the readability and overall presentation. A careful language edit by a native or professional academic editor is recommended prior to publication.

Author Response

Comments and Suggestions for Authors

This scoping review provides a well-structured and timely overview of the relationship between vitamin D deficiency and gestational diabetes mellitus (GDM) in Western countries. The manuscript demonstrates a strong understanding of the topic and effectively synthesizes evidence from diverse sources. The thematic categorization of results—particularly around maternal age, ethnicity, BMI, and seasonality—adds clarity and relevance to the discussion.

Response: We thank the reviewer for such positive feedback.

However, several areas could benefit from improvement:

  1. Language and Clarity: The overall readability would be enhanced by simplifying complex sentences and removing redundant expressions. Minor grammatical adjustments are also recommended throughout the manuscript to improve clarity and flow.

Response: Thank you for this constructive comment. The paper has been revised for language and grammar

  1. Methods Section: Although the methodology follows the JBI and PRISMA-ScR frameworks appropriately, certain parts (e.g., keyword strategy and exclusion reasons) should be more precisely and concisely described to increase transparency and reproducibility.

Response: We appreciate the reviewer’s feedback and agree that greater precision improves transparency and reproducibility. In response, we have expanded the Methodology section to clearly outline the keyword strategy, including a list of core search terms and Boolean operators. We have also clarified the predefined exclusion criteria used during screening. These refinements better reflect the rigour of the JBI and PRISMA-ScR frameworks followed in this review.

This is how the newly improved section read

“A structured search strategy was developed based on the JBI guidelines and the PCC framework (Population–Concept–Context). The following keywords and Boolean operators were used across all databases:

  • ("vitamin D deficiency" OR "25(OH)D" OR "hypovitaminosis D") AND
  • ("gestational diabetes" OR "GDM") AND
  • ("Western countries" OR "Australia" OR "Canada" OR "United States" OR "Europe").

Synonyms and Medical Subject Headings (MeSH) terms were adapted for each database. The search was conducted on March 10, 2025, across ProQuest Public Health, Google Scholar, BMC Public Health, and Scopus.

Eligibility Criteria:
Inclusion criteria:

  • Studies conducted in Western countries
  • Peer-reviewed articles examining the relationship between vitamin D deficiency and GDM
  • Observational or interventional designs
  • Published in English

Exclusion criteria:

  • Studies involving non-pregnant populations or non-Western settings
  • Reviews without original data
  • Studies lacking serum 25(OH)D measures or GDM outcomes
  • Conference abstracts, editorials, and grey literature”
  1. Discussion and Interpretation:
    • While the discussion effectively highlights key mechanisms and moderators, additional elaboration on inconsistent findings (e.g., null associations) would be helpful.

Response: We appreciate the reviewer’s insightful comment. In response, we have expanded the discussion to more directly reflect on studies that reported null or inconsistent associations between vitamin D deficiency and GDM. We acknowledge possible methodological and contextual explanations for these discrepancies, including differences in timing of vitamin D measurement, supplementation regimens, sample size, and diagnostic criteria. This addition provides a more balanced synthesis of the evidence and strengthens the interpretative rigour of the review. This is how the discussion section concerning the above comment reads now:

“However, findings across studies remain mixed. For instance, Zhang et al. (2008) reported a 3.7-fold increased risk of GDM among vitamin D-deficient women, whereas Clifton-Bligh et al. (2008) found no statistically significant association. These discrepancies may stem from variations in study design, such as small sample sizes, differing gestational timing of serum 25(OH)D measurement, or vitamin D supplementation protocols. Additionally, heterogeneity in GDM diagnostic criteria and unmeasured confounders (e.g., dietary intake, sun exposure) may have influenced effect estimates. These inconsistencies highlight the need for more standardised methodologies and underscore the importance of caution in drawing causal inferences from observational studies.

    • The mention of COVID-19 is interesting but would be stronger with clarification on whether included studies were pre- or post-pandemic.

Response: A statement was added to the COVID section highlighting studies done post-COVID- thank you

  1. Limitations Section: Please revise ambiguous expressions such as “while this may be the case…” to clearer, grammatically complete sentences. Also consider elaborating slightly on the potential impact of excluding grey literature and non-English studies.

Response: please find improved limitation section: “This review is limited by its geographic focus on Western countries, which excluded potentially relevant evidence from high-GDM-burden regions such as Asia and the Middle East. We acknowledge that patterns of vitamin D deficiency and GDM may differ significantly in non-Western settings due to environmental, cultural, and healthcare system factors. Comparative analyses across diverse regions would offer valuable insights for future global research.

There was notable heterogeneity among the included studies in terms of study design, population characteristics, diagnostic criteria for GDM, and methods of measuring vitamin D status. The predominance of observational studies further limits causal interpretation. Nevertheless, the findings highlight associations that warrant further exploration, particularly given the feasibility and low risk of vitamin D–based preventive strategies.

Finally, language restrictions and the exclusion of grey literature may have led to the omission of relevant studies, potentially limiting the comprehensiveness of the evidence base.”

  1. Recommendations and Conclusion: These sections are well written, but could benefit from a more explicit link between your findings and the proposed clinical/public health actions (e.g., vitamin D screening policies).

Response: that’s a really constructive comment, and while our original Recommendations and Conclusion sections are already strong, to satisfy the reviewer fully, we have just tightened the link between the findings and actionable public health/clinical strategies, especially around screening and policy. Please see the improved sections in the paper.

Comments on the Quality of English Language

The manuscript is generally well-organized and readable; however, several sentences are overly long or contain redundant phrasing, which occasionally hinders clarity. Minor grammatical errors and awkward constructions appear throughout the text, particularly in the Methods and Discussion sections. Simplifying sentence structure, ensuring consistent terminology, and improving transitions between ideas will significantly enhance the readability and overall presentation. A careful language edit by a native or professional academic editor is recommended prior to publication.

Response: Thank you for the constructive comment- please see the newly improved sections throughout the paper.

Round 2

Reviewer 2 Report

Comments and Suggestions for Authors

Accept in present form

Author Response

we have responded to all reviewers comments- please let us know if anything is missing.

thank you,

Fahad, on behalf of the authors